# By-Products of the Black Soybean Sauce Manufacturing Process as Potential Antioxidant and Anti-Inflammatory Materials for Use as Functional Foods

**DOI:** 10.3390/plants10122579

**Published:** 2021-11-25

**Authors:** Shu-Ling Hsieh, Yi-Wen Shih, Ying-Ming Chiu, Shao-Feng Tseng, Chien-Chun Li, Chih-Chung Wu

**Affiliations:** 1Department of Seafood Science, National Kaohsiung University of Science and Technology, Kaohsiung 81157, Taiwan; slhsieh@nkust.edu.tw; 2Department of Food and Nutrition, Providence University, Taichung 43301, Taiwan; a0922287120@gmail.com; 3Department of Allergy, Immunology, and Rheumatology, Tungs’ Taichung Metro Harbor Hospital, Taichung 43503, Taiwan; ymcgreen@yahoo.com.tw; 4Department of Quality Control and Research, Ta-Tung Soya Sauce Co. Ltd., Yunlin 64069, Taiwan; blackbeanbeauty@gmail.com; 5Department of Nutrition, Chung Shan Medical University, Taichung 40201, Taiwan; chienchien367@gmail.com

**Keywords:** black bean soybean sauce, by-product, antioxidant, inflammation, functional supplements

## Abstract

To assess the potential of by-products of the black bean fermented soybean sauce manufacturing process as new functional food materials, we prepared black bean steamed liquid lyophilized product (BBSLP) and analysed its antioxidant effects in vitro. RAW264.7 macrophages were cultured and treated with BBSLP for 24 h, and 1 μg/mL lipopolysaccharide (LPS) was then used for another 24 h to induce inflammation. The cellular antioxidant capacity and inflammatory response were then analysed. Activation of nuclear factor kappa B (NF-κB) signaling in RAW264.7 macrophages was also analysed. Results showed BBSLP had 2,2-diphenyl-1-(2,4,6-trinitrophenyl)hydrazyl (DPPH) and 2,2′-azino-bis(3-ethylbenzothiazoline-6-sulfonic acid) diammonium (ABTS^+^) radical-scavenging abilities and reducing power in vitro. The levels of both reactive oxygen species (ROS) and thiobarbituric acid reactive substances (TBARS) were reduced after RAW264.7 macrophages were treated with BBSLP after LPS induction. After RAW264.7 macrophage treatment with BBSLP and induction by LPS, the levels of inflammatory molecules, including nitric oxide (NO), prostaglandin E_2_ (PGE_2_), IL-1α, IL-6 and TNF-α, decreased. NF-κB signaling activity was inhibited by reductions in IκB phosphorylation and NF-κB DNA-binding activity after RAW264.7 macrophages were treated with BBSLP after LPS induction. In conclusion, BBSLP has antioxidant and anti-inflammatory capabilities and can be a supplement material for functional food.

## 1. Introduction

Soybean sauces are commonly used as seasonings and sauces in Asia. Among them, black bean-fermented soybean sauce which uses black beans (*Glycine max* (L.) Merr.) as the major raw material is one of the most favored. During the soybean sauce manufacturing process, raw soybean steaming is an important step before yeast inoculation [1]. However, steamed soybean liquid waste, particularly by-products, may contain nutrients and other phytochemicals. These by-products may not only provide nutrient and active compounds for supplements but also possess sustainable and circular economic characteristics if they can be developed into a healthy food material. Considering global sustainability, the circular economy is a new alternative approach to the traditional economy [2]. Therefore, recycling certain by-products from food manufacturing processes as new food and pharmaceutical materials is not only a solution for food waste and supply issues, but also a new preferred resource for human health [2].

The black soybean cultivar has abundant polyphenols in its seed coating [3]. During the black soybean sauce manufacturing process, the beans are steamed at a high temperature and pressure [4]. This process is similar to soybean extraction with high-temperature steam under high pressure. These steamed soybean extracts contain polyphenols and some of the active components of black soybeans [5]. Previous studies have shown that black soybeans can reduce cardiovascular disease [6], regulate blood sugar [7] and have anticancer effects, [8,9], improve bone resorption in menopause [10], and possess antioxidative [11] and anti-inflammatory [12] properties. Kim et al. [13] showed that raw black soybeans have abundant total free polyphenols, flavonoids and phenolic acids. Anthocyanidin is a type of isoflavone that is present in black soybean coats [9]. Recycling this black soybean steamed liquid (BBSL) has the potential to providing an effective physiological material for functional food development.

The inflammatory response is a physiological protective mechanism of the body that occurs in response to infection; however, the occurrence of chronic inflammation with chronic inflammatory cells, including macrophages, lymphocytes and plasma cells, among others, is increasing [14]. During the inflammatory response process, reactive oxygen species (ROS), nitric oxide (NO) and prostaglandin E_2_ (PGE_2_) act as messengers for different physiological functions and pathological processes [15]. Conversely, intracellular cytokines, such as IL-1β, IL-6, IL-10 and TNF-α, are secreted from macrophages and lymphocytes [16]. These cytokines mediate the immune response and influence the macrophage microenvironment [17]. Excess or long-term chronic inflammation will lead to chronic diseases, such as cardiovascular disease (CVD), sepsis, diabetes mellitus (DM), and chronic kidney disease (CKD), increasing human health risks [18].

Some inflammatory response factors, such as IL-1β, IL-6, and TNF-α, and inflammation-mediated molecular enzymes, including inducible nitroxide synthase (iNOS) and cyclooxygenase-2 (COX-2), are primarily controlled by nuclear factor kappa B (NF-κB), which is well recognized to play an important role in inflammation [15,19,20,21,22].

Currently, the food supply chain has many challenges due to decreased natural resources and increased food waste [23]. Thus, recycling reusable by-products of plant foods manufacturing process as new foods or components with active physiological effects is a forward-looking issue. For further more study and assessment the potential of this by-product of the soybean sauce manufacturing process on functional food material. The present study aims to investigate the capability of antioxidation and anti-inflammation of black bean steamed liquid lyophilized product (BBSLP).

## 2. Results

### 2.1. pH Value, Flavonoids, Total Phenol, and Protein Contents in BBSL and BBSLP

To know the application of fresh BBSL, the pH was measured in this study. As shown in Table 1, the pH of fresh BBSL was 5.84 ± 0.12. The flavonoids, total phenol, and protein contents of BBSL were 11.5 ± 1.5 mg rutin equivalents (RUE)/mL, 3.83 ± 0.3 mg gallic acid equivalents (GAE)/mL and 3.5 ± 0.8 mg/mL, respectively (Table 1). In addition, after fresh BBSL was concentrated by a rotary vacuum dryer and frozen dry by a frozen dryer, the flavonoids, total phenol, and protein contents of BBSLP were 0.1 ± 0.01 mg RUE/mg, 0.03 ± 0.01 mg GAE/mg and 0.31 ± 0.04 mg/mg, respectively (Table 1).

### 2.2. BBSLP Showed Antioxidant Effects In Vitro Conditions

In the 2,2-diphenyl-1-(2,4,6-trinitrophenyl)hydrazyl (DPPH) radical-reducing ability test the scavenging abilities of 0.5, 1 and 2 mg/mL BBSLP were 42.5 ± 8.7%, 54.4 ± 0.4% and 79.6 ± 0.2%, respectively. The observed scavenging ability in the vitamin C-treated group was 88.0 ± 5.8% (Figure 1A). Although the DPPH radical-reducing ability in the BBSLP groups was lower than that in the vitamin C group, it increased in a dose-dependent manner.

The EC_50_ value of BBSLP for DPPH radical-scavenging ability was 0.81 mg/mL. Figure 1B shows that the Trolox-treated group demonstrated 95.5 ± 4.3% 2,2′-azino-bis(3-ethylbenzothiazoline-6-sulfonic acid) diammonium (ABTS^+^) radical scavenging. The ABTS^+^ radical scavenging abilities in the 0.5, 1 and 2 mg/mL BBSL groups were 7.8 ± 2.3%, 20.5 ± 6.3% and 50.2 ± 8.3%, respectively, and showed a dose-dependent increase (*p* < 0.05). The effect concentration (EC_50_) value of BBSLP for ABTS^+^ radical scavenging ability was 1.50 mg/mL. In addition, the reducing power of 0.5, 1 and 2 mg/mL BBSLP reached 42.5 ± 84.6% in a dose-dependent manner (*p* < 0.05). The vitamin C-treated group showed 88.7 ± 9.6% reducing power (Figure 1C). The EC_50_ value of BBSLP for reducing power was 0.80 mg/mL.

### 2.3. BBSLP Maintained the Viability of RAW264.7 Macrophages after Lipopolysaccharide (LPS) Induction

Our preliminary experimental results showed that the viability of RAW264.7 macrophages treated with 0.1 to 5 μg/mL BBSLP was not significantly different compared with that of the control cells (data not shown). For reasons related to BBSLP yield and solubility, we used 0.1, 0.5 and 1 μg/mL BBSLP as the experimental doses in the following study.

In the present study, the cell viability of RAW264.7 macrophages did not significantly differ between the 0.1, 0.5 or 1 μg/mL BBSLP (approximately 97–101%) under LPS induction or the control group (100%) (Figure 2A). Based on morphological examination results using inverted microscopy (Figure 2B), the cell number and morphology did not significantly differ between any BBSLP-treated group under LPS induction and the control group. Under LPS induction, RAW264.7 macrophage treatment with 0.1, 0.5, or 1 μg/mL BBSLP did not affect viability.

### 2.4. BBSLP Reduced Oxidative Stress in RAW264.7 Macrophages after LPS Induction

When RAW264.7 macrophages were treated with LPS alone, the thiobarbituric acid reactive substances (TBARS) level was significantly increased by 229% (*p* < 0.05) (Figure 3A). However, when RAW264.7 macrophages were treated with 0.1, 0.5 or 1 μg/mL BBSLP and then stimulated with LPS, the TBARS levels were significantly decreased by 38 to 70% compared with those in the LPS alone-treated group (*p* < 0.05) (Figure 3A). The TBARS levels in the group treated with only 5 μg/mL BBSLP did not differ from those in the control group. The ROS levels (100%) of RAW264.7 macrophages treated with only LPS were significantly higher than those in control cells (23.1 ± 5.3%) (*p* < 0.05) (Figure 3B). However, the ROS levels in RAW264.7 macrophages did significantly decrease in the groups treated with 0.1, 0.5 or 1 μg/mL BBSLP after stimulation with LPS (approximately 21.2–66.5%) (*p* < 0.05).

### 2.5. BBSLP Reduced NO and PGE_2_ Production in RAW264.7 Macrophages after LPS Induction

NO and PGE_2_ production was significantly increased after RAW264.7 macrophages were induced with LPS compared with the control group (*p* < 0.05, Figure 4A,B). However, when RAW264.7 macrophages were cotreated with 1 µg/mL BBSLP and LPS, NO levels were decreased by 15% compared with those in the LPS group (*p* < 0.05). A 60–68% decrease in PGE_2_ was noted in cells cotreated with 0.1 to 1 μg/mL BBSLP and LPS (*p* < 0.05). Immunoblot analysis showed that the iNOS levels in RAW264.7 macrophages treated with 0.1, 0.5 and 1 μg/mL BBSLP were 82%, 77% and 69% that of cells treated with LPS alone, respectively (*p* < 0.05, Figure 4C,D). When RAW264.7 macrophages were treated with 0.1, 0.5 and 1 μg/mL BBSLP, the COX-2 protein levels were significantly reduced by 84%, 84% and 82%, respectively, compared with those in the LPS-treated group (*p* < 0.05, Figure 4C,D).

### 2.6. BBSLP Decreased IL-1β, IL-6 and TNF-α Levels in RAW264.7 Macrophages after LPS Induction

Figure 5A shows that the IL-1β level was significantly increased after an inflammatory response was induced in RAW264.7 macrophages by LPS compared with the control group (*p* < 0.05); however, when RAW264.7 macrophages were treated with 0.5 or 1 μg/mL BBSLP combined with LPS, the IL-1β levels were 59.9 ± 8.7% and 32.0 ± 8.7%, they were significantly lower than the LPS induction alone group (100%, *p* < 0.05, Figure 5A). The IL-6 levels in RAW264.7 cells decreased significantly by 7% after treatment with 1 μg/mL BBSLP compared with those after LPS induction alone (*p* < 0.05, Figure 5B). Figure 5C also shows that the TNF-α level in the group treated with only 1 μg/mL BBSLP decreased significantly compared with the LPS alone group (*p* < 0.05, Figure 5B). Notably, IL-10 production did not differ among the control group, LPS-treated group, the group treated with various concentrations of BBSLP combined with LPS, and the group treated with BBSLP alone (Figure 5D).

### 2.7. BBSLP Reduced the Activation of NF-ĸB Signalling in RAW264.7 Cells after LPS Induction

Figure 6A,B show that IκB phosphorylation was significantly reduced by 20–50% after 0, 1, 0.5 or 1 μg/mL BBSLP treatment in RAW264.7 cells induced with LPS (*p* < 0.05). However, BBSLP did not affect the protein contents of cytosolic I-κB in EA.hy926 cells (Figure 6A,B). The nuclear NF-κB levels were significantly decreased by 24% and 16%, respectively, after 0.5 or 1 µg/mL BBELP treatment in RAW264.7 cells induced by LPS (*p* < 0.05, Figure 6A,B). The DNA-binding activity of nuclear NF-κB was significantly suppressed by 49% in cells treated with 100 µg/mL BBSLP (Figure 6C).

## 3. Discussion

The present study showed that BBSLP has the ability of free radical scavage and reducing power enhance in vitro. And, the potential antioxidant and anti-inflammatory effects of BBSLP in LPS-induced RAW264.7 macrophages. Because BBSLP significantly reduced oxidative stress, including an ability to decrease levels of free radicals and lipid peroxidation, and reduced pro-inflammation molecules, including IL-1β, IL-6 and TNF-α levels, in LPS-induced RAW264.7 cells. 

Our results found that BBSLP contained flavonoids and polyphenols, which were may be involved in its high antioxidative and anti-inflammatory properties. Glevitzky et al. [24] showed that there are a high intercorrelation between the number of phenolic groups within the basic structure of flavonoids and their antioxidant activity and between the antioxidant activity and the number of -OH phenolic groups also with a high correlation. However, polyphenols or the other components of BBSL and BBSLP whether playing a major or important role in antioxidation or not, are still need furthermore composition analysis and investigation. Inflammatory response mediators, including PGE_2_ and NO, and inflammatory cytokines, including IL-1β, IL-6, IL-10 and TNF-α, were all regulated, reducing inflammation in LPS-induced RAW264.7 cells. Furthermore, BBSLP could downregulate NF-κB signalling activation, which led to reduced iNOS, COX-2, IL-1β, IL-6, IL-10 and TNF-α transcription.

Oxidative stress usually triggers the inflammatory response in various cells and tissues [25]. Macrophages then use ROS production to scavenge xenobiotics, including bacteria and oxidized low-density lipoprotein (ox-LDL) [26]. Long-term inflammation, chronic inflammation and oxidative stress lead to chronic diseases, such as CKD, CVD, DM and sepsis with a poor prognosis. In the present study, BBSLP displayed excellent free radical-scavenging ability, reducing power and SOD activity in an in vitro model and significantly reduced ROS production and TBARS levels in LPS-induced RAW264.7 cells. The above excellent antioxidative effects were from the polyphenols and isoflavones contained in BBSLP. Takahashi et al. [27] showed that the seed coats of black soybeans have a higher total polyphenol content than those of yellow soybeans. Black soybeans may more effectively inhibit LDL oxidation than yellow soybeans because of the higher total polyphenol contents in their seed coat. Additionally, cyanidin-3-glucoside, petunidin-3-glucoside and peonidin-3-glucoside, three major anthocyanins, have been detected in black soybean seed coats [3]. DPPH radical scavenging, ABTS^+^ radical scavenging and ferric reducing antioxidant power (FRAP) analysis results have also shown that the black soybean seed coat is a more efficient reducing agent than dehulled black soybeans and yellow soybean coats [3]. In addition, black soybeans are rich in polyphenols, including isoflavones, anthocyanidins and flavan-3-ols. Moreover, black soybeans can prevent CVD risks by increasing polyphenol concentrations and decreasing oxidative stress in healthy women [28]. Previous studies in different experimental models have shown that polyphenols and isoflavones also have anti-inflammatory characteristics. Takekawa et al. [29] showed that genistein, a soybean polyphenol, can significantly suppress water immersion restraint (WIR) stress-induced gastric mucosal injury. The underlying mechanism involves a significant elevation of SOD activity and significant suppression of both TBARS levels and the production of TNF-α to protect against gastric mucosal injury [29]. Additionally, puerarin, an isoflavonoid extracted from Kudzu roots, reduces malondialdehyde levels, increases SOD activity and alleviates TNF-α, IL-1β and IL-6 protein levels in the hippocampus. Antioxidation and anti-inflammation are induced by the streptozotocin (STZ) group to protect DM rats from cognitive deficits [30].

In addition, resveratrol, a polyphenol constituent of grapes, acts as a COX suppressor, reducing the inflammatory response similarly to a nonsteroidal anti-inflammatory drug (NSAID). A molecular basis for the mutually beneficial relationship between plants and humans has been speculated [31]. Hussain et al. [32] reported the anti-inflammatory and antioxidative properties of polyphenols, the mechanisms by which polyphenols inhibit molecular signalling pathways that have been activated by oxidative stress, and the roles of polyphenols in inflammation-mediated chronic disorders. The above data and previous reports indicate that BBSLP, an extract rich in polyphenols and isoflavones, can significantly reduce oxidative stress, inhibit IL-1β, IL-6 and TNF-α and increase IL-10 as an anti-inflammatory material in RAW264.7 macrophages. BBSLP may be helpful for the development of future antioxidant therapeutics and new anti-inflammatory drugs [33].

In the present study, BBSLP significantly inhibited NF-κB signalling activity by reducing I-κB phosphorylation and NF-κB-DNA binding activity. NF-κB signalling plays an important role in iNOS, COX-2, IL-1β, IL-6 and TNF-α transcription [15,19,20,21,22]. Previous studies have shown that one of the major mechanisms of reducing inflammation is to reduce NF-κB signalling activity and the expression of proinflammatory molecules [23]. Bao et al. [34] showed that chlorogenic acid, a major polyphenol compound from coffee, can prevent diabetic nephropathy by inhibiting oxidative stress and inflammation through the reduction of NF-κB signalling activity. Singh et al. [35] also reported that polyphenols have antioxidative and anti-neuroinflammatory properties by regulating NF-κB activation in neurodegenerative diseases.

Currently, the food supply chain is facing substantial pressures, including the availability of fewer natural resources and increased food waste [23]. One important way to increase the food supply and decrease the environmental consequences of current food production is to reduce food waste levels and their economic, environmental and social implications [36]. Previous studies have shown that various by-products from the manufacturing of animals and plants for food contain various fatty acids [37], phytochemicals [38,39], and amino acids [40], all of which are beneficial to the food supply chain and/or can act as health promoters for food sustainability. In the present study, BBSLP showed potential antioxidant and anti-inflammatory effects. In the present study, BBSLP showed preliminary potential on antioxidant and anti-inflammatory effects in vitro. However, the molecular mechanisms of the physiological effects require further study in animal models or human clinical trials. On the other hand, how does BBSLP apply in function foods? Where BBSLP to be developed into a functional food material, a functional assessment of this product is needed. The above questions are an important issue for BBSLP application.

## 4. Materials and Methods

### 4.1. Materials

BBSL was a gift from Ta-Tung Soya Sauce Co., Ltd., located in Siluo Town (Yunlin, Taiwan). BBSL is collected from the soybean sauce manufacturing process. After fresh black beans are washed and steamed by 120 °C streams in a closed steam tank for 1 h. BBSL were directly collected as experimental materials. LPS was purchased from Sigma-Aldrich Co. (St. Louis, MO, USA).

### 4.2. Preparation of BBSLP

According to our previous methods [41], BBSLP was prepared by our laboratory. Fresh BBSL was concentrated in a rotary evaporator (N-1110, Tokyo Rikakikai Co., Ltd., Tokyo, Japan) and then dried in a freeze dryer (Freezone 4.5, Labconco, Kansas City, MO, USA) at −43 °C. The BBSLP was stored at −20 °C until use. The percent yield of BBSLP was 1.16% (*w*/*v*).

### 4.3. Determination of the pH Value and Total Flavonoids, Phenols and Protein in BBSL and BBSLP

The pH values of fresh BBSL were measured using a pH metre (MP220 pH meter, Mettler Toledo, Greifensee, Switzerland). The total phenol contents were analysed using a colorimetric method according to Padmavati et al. [42]. One hundred microlitres of 1 N Folin-Ciocalteu reagent (Sigma-Aldrich Co.) was added to 100 μL of diluted BBSL or BBSLP (dissolved in reverse-osmosis (RO) H_2_O). Then, 500 µL of 7.5% Na_2_CO_3_ solution was added to react for 30 min, and the absorbance of each sample was measured at an optical density (OD) of 760 nm in a Biokinetics microplate reader (Bio-Tek Instruments, Winooski, VT, USA). Calibration curves were constructed using 0, 0.125, 0.25 and 0.5 mg/mL gallic acid (GA). The total phenolic content is represented as mg GAE/mL BBSL or mg GAE/g BBSLP.

The flavonoid content was analysed using the colorimetric method according to Jia et al. [43]. One hundred microlitres of 5% NaNO_3_ solution was added to 100 μL of the BBSL or BBSLP solution to react for 5 min. Then, 50 μL of 10% AlCl_3_ solution was added. Finally, 600 μL of 4% NaOH solution was added to the mixture for 30 min. The absorbance of the mixture was measured at OD 510 nm on the Biokinetics microplate reader. Calibration curves were constructed with 0, 0.02, 0.06, 0.08 and 0.1 mg/mL rutin (RU) as the standard. The total flavonoid content is represented as mg RUE/mL BBSL or mg RUE/g BBSLP.

Crude protein contents were analysed according to Lowry et al. [44]. Fifty microlitres of each standard of BBSL or BBSLP was added to 50 μL of trichloroacetic acid and standing for reacted for 30 min at room temperature. The mixture was centrifuged at 15,000× *g* for 20 min at 4 °C. Then, the supernatant was discarded. The precipitate was dissolved in 1 mL of NaOH and standing for reacted for 30 min at room temperature. Then, 1.0 mL of modified Lowry Reagent (Sigma-Aldrich Co.) was added, then mixing and incubation at room temperature for 10 min. Five hundred µL Prepared 1X Folin-Ciocalteu’s phenol reagent (Sigma-Aldrich Co.) was added, and the mixture was standing in a water bath at 37 °C for 30 min. After 30 min, the absorbance was measured at 660 nm on the Biokinetics microplate reader.

### 4.4. In Vitro Antioxidant Ability of BBSLP

In this study, DPPH (Sigma-Aldrich Co.) radical scavenging activity by BBSLP was analysed according to the method of Shimada et al. [45]. For the measure of the DPPH radical-scavenging activity, 1.5 mL of the sample solution with varying BBSLP concentrations (0.5, 1, 2 and 4 mg/mL) were added 1.5 mL of 0.15 mM DPPH in 50% ethanol. The mixture was mixed and incubated at room temperature in the dark for 30 min. The optical density at 517 nm was measured using the Biokinetics microplate reader. In this test, 1 mg/mL vitamin C was used as a control. The scavenging activity was calculated as (1 − A_BBSLP_ or A_vitamin C_/A_blank_) × 100.

According to the method of Shimada et al. [45] to analyse the reducing power activity of BBSLP in vitro. Here, 0.5 mL of the 0.5, 1, 2 and 4 mg/mL BBSLP, respectively were mixed with 2.5 mL of 0.2 M phosphate buffer (pH 6.6) and 2.5 mL of 1% potassium ferricyanide, then the mixture was incubated at 50 °C for 20 min. A 2.5 mL aliquot of 10% trichloroacetic acid was added to the mixture, and the mixture was then centrifuged at 3000× *g* for 10 min. The supernatant (2.5 mL) was mixed with 2.5 mL of distilled water and 2.5 mL of 0.1% ferric chloride, and the absorbance at 700 nm was read using the microplate reader. The reducing power was calculated as (A_BBSLP_ or A_vitamin C_ − A_blank_)/A_vitamin C_ × 100. A vitamin C (1 mg/mL) was used as a control.

The ABTS^+^ radical scavenging ability of 0.5, 1, 2 or 4 mg/mL BBSLP was analysed according to the method described by Re et al. [46]. Use a 10 μL of the sample solution with varying BBSLP concentrations (0.5, 1, 2 and 4 mg/mL) were added 990 μL of 2 mM 2,2′-azino-bis(3-ethylbenzothiazoline-6-sulfonate) radical cation (ABTS^•+^) solution. The mixture was mixed and incubated at room temperature in the dark for 10 min. The optical density at 737 nm was measured using the microplate reader. The ABTS^+^ radical-scavenging ability was calculated as (A_blank_ or A_BBSLP_ − A_Trolox_)/A_blank_ × 100. In this test, 1 mg/mL Trolox was used as a control.

### 4.5. Cell Culture and Treatment

RAW264.7 macrophages and mouse monocyte macrophages were purchased from the Bioresource Collection and Research Center (Hsinchu, Taiwan). Dulbecco’s modified Eagle’s medium containing 42 mM L-glutamine, 100 units/mL penicillin, 100 μg/mL streptomycin and 10% (*v/v*) heat-inactivated fetal bovine serum (FBS; Gibco, Thermo Fisher Scientific, Inc., Waltham, MA, USA) was used as the culture medium. All cultured cells were incubated in an atmosphere of 5% CO_2_/95% air at 37 °C.

In this study, 1 × 10^5^ RAW264.7 macrophages per 30 mm plate or 1 × 10^6^ per 60 mm plate were cultured for various biochemical tests. RAW264.7 macrophages were incubated with 0.5, 1 or 5 μg/mL BBSLP for 24 h and then induced with 1 μg/mL LPS (Sigma-Aldrich Co.) for another 24 h. LPS was used to induce inflammation [47]. The induced control group was treated with 1 μg/mL LPS alone. BBSLP was soluble in sterilized H_2_O, and cells treated with sterilized H_2_O alone served as the control group. BBSLP (1 µg/mL) without LPS treatment for 48 h made up another control group.

### 4.6. Cell Viability Analysis

To determine the optimum test concentration of BBSLP for use in this study, the cell viability of RAW264.7 macrophages was analysed according to the method of Denizot and Lang [48]. After RAW264.7 macrophages were incubated in DMEM containing 0.5 mg/mL thiazolyl blue formazan (MTT; Sigma-Aldrich Co.) for an additional 3 h, the medium was removed and extracted with isopropanol for 15 min. The isopropanol fraction was measured with the Biokinetics microplate reader at OD 570 nm. To evaluate morphological changes, a phase-contrast inverted fluorescence microscope (Olympus IX51, Olympus, Tokyo, Japan) was used.

### 4.7. Measurement of Lipid Peroxidation and ROS Levels

The effect of BBSLP on lipid peroxidation in RAW264.7 macrophages induced by LPS was determined according to the method of Fraga et al. [49]. The lipid peroxidation indicator thiobarbituric acid reactive substances (TBARS) was extracted and measured with a fluorescence microplate reader (excitation wavelength 515 nm and emission wavelength 555 nm, Bio-Tek Instruments, Winooski, VT, USA). The protein levels were determined according to the method described by Lowry et al. [44]. The TBARS level is shown in nmol TBARs/mg protein. The levels of ROS in RAW364.7 cells were determined using a Cellular ROS Assay Kit (ab113851, Abcam Inc., Cambridge, MA, USA).

### 4.8. Determination of Nitrite (NO) and Prostaglandin E_2_ (PGE_2_)

To determine the inflammation level in RAW 264.7 cells, the Griess assay was used [50]. Griess reagent (1% sulfanilamide/0.1% N-(1-naphtyl)ethylene diamine dihydrochloride in 2.5% H_3_PO_4_) was mixed with an equal part of the cell culture medium of control or various experimental groups RAW 264.7 cells. In this test, the NO_2_ content was used as an indicator of NO content in RAW364.7 macrophages. The OD 550 nm was determined and calibrated using a standard curve of NaNO_2_ prepared in culture medium.

The PGE_2_ levels in RAW264.7 macrophages were determined using an enzyme immunoassay (EIA) kit (ADI-900-001, Cayman Chemical, Ann Arbor, MI, USA). The cell culture supernatants were collected after experimental treatment and centrifuged at 1000× *g* for 15 min to remove the particulate matter. The medium and PGE2 EIA conjugate was added to a 96-well plate pre-coated with goat anti-mouse IgG and left to react for 1 h, followed by a final wash to remove any unbound antibody-enzyme reagent. A substrate solution was added and the intensity of the color produced was measured at 412 nm. The concentration of PGE_2_ in each sample was calculated according to PGE_2_ standards.

### 4.9. Measurement of IL-1β, IL-6 and TNF-α

The levels of IL-1β, IL-6 and TNF-α in RAW264.7 macrophages were analysed using rat IL-1β/IL-1F2 DuoSet ELISA (R&D, DY501-05), rat IL-6 DuoSet ELISA (R&D, DY506-05), and rat TNF-α DuoSet ELISA (R&D, DY510-05) kits (R&D Systems, Inc., Minneapolis, MN, USA), respectively, according to the manufacturer’s instructions. In brief, capture antibodies, cultured medium supernatants, detection antibodies, streptavidin-conjugated horseradish-peroxidase were processed on the plate in order, and the color subtract tetramethylbenzidine was used. The absorbance was measured and the concentration was calculated according to the standard.

### 4.10. Immunoblot Analyses of iNOS, COX-2 and NF-κB Signalling Molecule Expression

The iNOS, COX-2 and NF-κB signalling molecule expression was analysed using the method described by Hsieh et al. [51]. At the end of the treatment, the cells were collected in 200 μL of lysis buffer (10 mM Tris-HCl, 5 mM EDTA, 0.2 mM phenylmethylsulfonyl fluoride and 20 μg/mL aprotinin, pH 7.4), and the protein content was determined according to the method of Lowry et al. [44].

Equal amounts (approximately 10–20 μg per sample) of cellular protein were separated by 10% sodium dodecyl sulfate (SDS) polyacrylamide gel electrophoresis (PAGE) [52], after which the samples were transferred to polyvinylidene difluoride (PVDF) membranes [53]. The PVDF membranes were then incubated with anti-iNOS, anti-COX-2, anti-p-IκB, anti-IκB, anti-NF-κB (p65) or anti-GAPDH antibodies at 4 °C overnight, followed by incubation with a peroxidase-conjugated secondary antibody. For density analysis, blots were treated with enhanced chemiluminescence substrate solutions and exposed using a ChemiDoc XRSt System (Bio-Rad Laboratories, Hercules, CA, USA). An NF-κB (p65) transcription factor activity assay kit (Cayman Chemical Co.) was used to analyse the NF-κB DNA binding activity of the nuclear fraction.

### 4.11. Statistical Analysis

The SPSS Statistical Analysis Software for Windows, version 20.0 (SPSS Inc., Chicago, IL, USA) was used to analyse the experimental data in the present study. One-way analysis of variance (ANOVA) and Duncan’s or Tukey’s multiple-range test were used to evaluate the significance of differences between each mean value. A *p*-value less than 0.05 was used to indicate a statistically significant result.

## 5. Conclusions

In conclusion, the presented data in this study demonstrate that BBSLP could reduce oxidative stress and pro-inflammation factors in LPS-induced RAW264.7 cells through the inhibition of the NF-κB signaling pathway, indicating that it may have potent antioxidant and anti-inflammatory capabilities, suggesting that BBSLP could be developed as a supplement material for functional foods.

## Figures and Tables

**Figure 1 plants-10-02579-f001:**
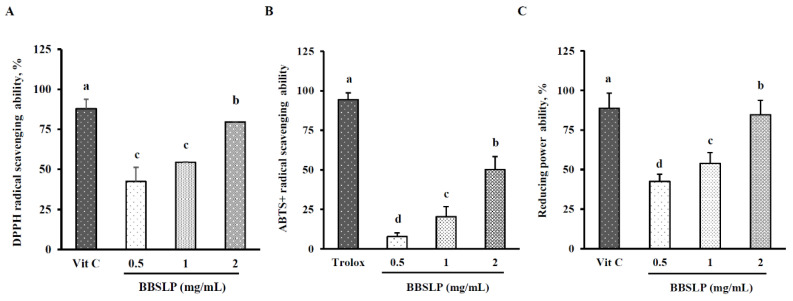
In vitro antioxidative ability of the BBSLP. DPPH radical scavenging activity (**A**), ABTS^+^ radical scavenging activity (**B**) and reducing power (**C**). Vitamin C was used as the positive control in the DPPH radical scavenging assay and reducing power ability assay, Trolox was used as the positive control in the ABTS^+^ radical scavenging. Values are presented as means ± SD (*n* = 3–5). ^abc^ Values are significantly different from the other groups as determined by Duncan’s test (*p* < 0.05). Black bean steamed liquid lyophilized product (BBSLP).

**Figure 2 plants-10-02579-f002:**
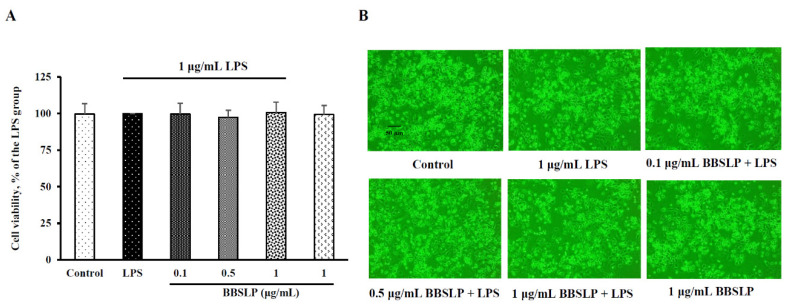
Effects of BBSLP on the viability of LPS-induced RAW264.7 cells. RAW264.7 cells (1 × 10^5^ cells/30-mm plate) were seeded and cultured overnight, treated with 0.1, 0.5 or 1 μg/mL BBSLP for 24 h and then induced or not with 1 μg/mL LPS for another 24 h. The group treated with 1 μg/mL LPS alone served as an induced control group. BBSLP was diluted in sterilized H_2_O, and cells treated with sterilized H_2_O alone served as the control group. Cells treated with 1 μg/mL BBSLP without LPS treatment for 48 h were used as another control group. Cell viability (**A**) and morphological changes (**B**) were examined. Values are presented as means ± SD (*n* = 3–5).

**Figure 3 plants-10-02579-f003:**
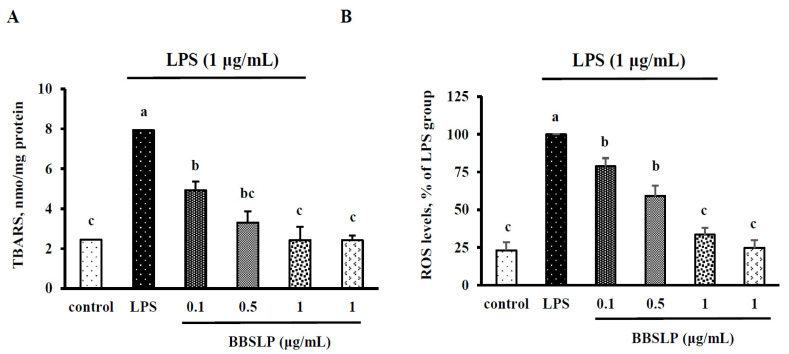
Effects of BBSLP on LPS-induced RAW264.7 cell oxidative stress. RAW264.7 cells (1 × 10^5^ cells/30-mm plate) were seeded and cultured overnight, treated with 0.1, 0.5 or 1 μg/mL BBSLP for 24 h and then induced or not with 1 μg/mL LPS for another 24 h. The group treated with 1 μg/mL LPS alone served as an induced control group. BBSLP was diluted in sterilized H_2_O, and cells treated with sterilized H_2_O alone served as the control group. Cells treated with 1 μg/mL BBSLP without LPS treatment for 48 h were used as another control group. TBARS levels (**A**) and ROS levels (**B**) were examined. Values are presented as means ± SD (*n* = 3–5). ^abc^ Values are significantly different from the other groups as determined by Tukey’s test (*p* < 0.05).

**Figure 4 plants-10-02579-f004:**
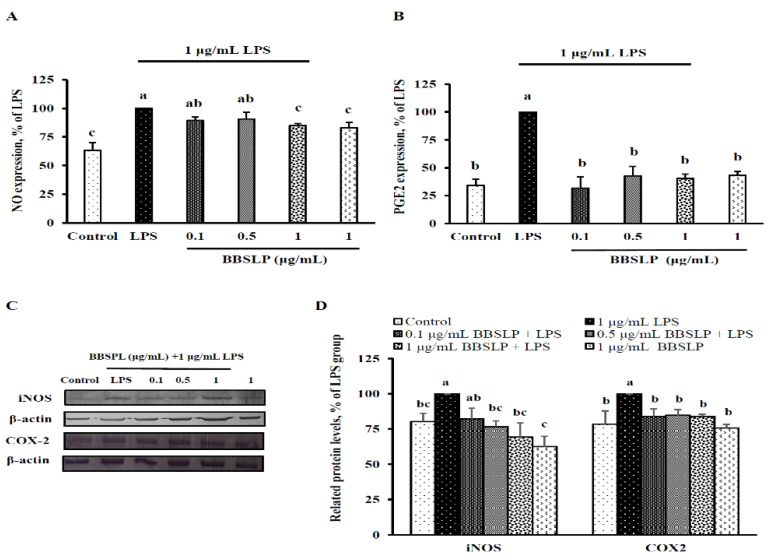
Effects of BBSLP on NO, PGE_2_, iNOS and COX-2 levels in RAW264.7 cells induced by LPS. RAW264.7 cells (1 × 10^5^ cells/30-mm plate) were seeded and cultured overnight, treated with 0.1, 0.5 or 1 μg/mL BBSLP for 24 h and then induced or not with 1 μg/mL LPS for another 24 h. The group treated with 1 μg/mL LPS alone served as an induced control group. BBSLP was diluted in sterilized H_2_O, and cells treated with sterilized H_2_O alone served as the control group. Cells treated with 1 μg/mL BBSLP without LPS treatment for 48 h were used as another control group. NO (**A**), PGE_2_ (**B**), iNOS and COX-2 (**C**) protein expression and quantified iNOS and COX-2 levels (**D**) were examined. Values are presented as means ± SD (*n* = 3–5). ^abc^ Values are significantly different from the other groups as determined by Tukey’s test (*p* < 0.05).

**Figure 5 plants-10-02579-f005:**
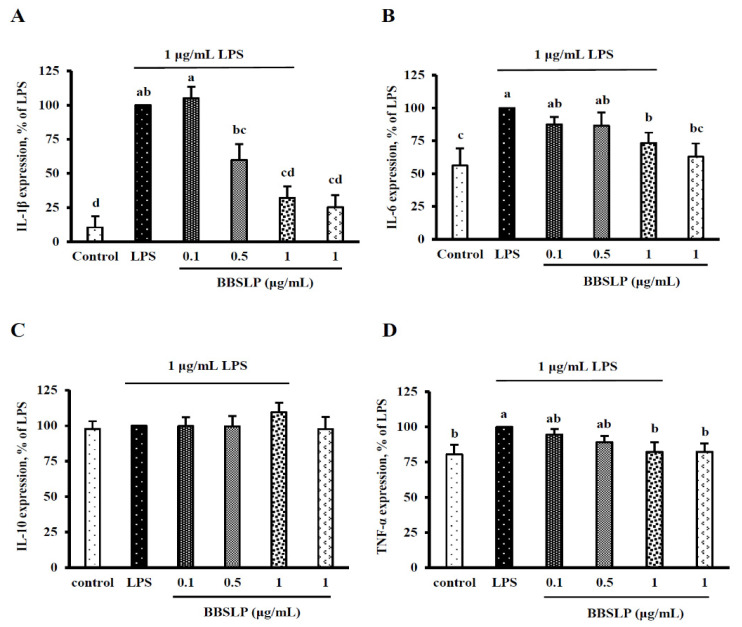
Effects of BBSLP on the inflammatory response in RAW264.7 cells induced by LPS. RAW264.7 cells (1 × 10^5^ cells/30-mm plate) were seeded and cultured overnight, treated with 0.1, 0.5 or 1 μg/mL BBSLP for 24 h and then induced or not with 1 μg/mL LPS for another 24 h. The group treated with 1 μg/mL LPS alone served as an induced control group. BBSLP was diluted in sterilized H_2_O, and cells treated with sterilized H_2_O alone served as the control group. Cells treated with 1 μg/mL BBSLP without LPS treatment for 48 h were used as another control group. Levels of NOIL-1β (**A**), IL-6 (**B**), IL-10 (**C**), and TNF-α (**D**) were examined. Values are presented as means ± SD (*n* = 3–5). ^abc^ Values are significantly different from the other groups as determined by Tukey’s test (*p* < 0.05).

**Figure 6 plants-10-02579-f006:**
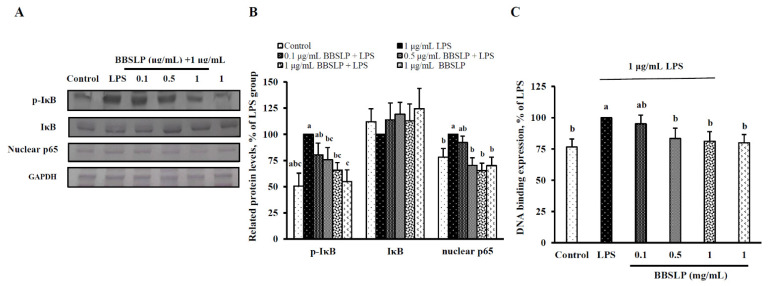
Effects of BBSLP on NF-κB signalling activation in LPS-induced RAW264.7 cells. RAW264.7 cells (1 × 10^5^ cells/30-mm plate) were seeded and cultured overnight, treated with 0.1, 0.5 or 1 μg/mL BBSLP for 24 h and then induced or not with 1 μg/mL LPS for another 24 h. The group treated with 1 μg/mL LPS alone served as an induced control group. BBSLP was diluted in sterilized H_2_O, and cells treated with sterilized H_2_O alone served as the control group. Cells treated with 1 μg/mL BBSLP without LPS treatment for 48 h were used as another control group. Phosphorylated IκB (p-IκB), IκB and nuclear p65 expression (**A**), quantified p-IκB, IκB and nuclear p65 levels (**B**) and NF-κB-DNA binding activity (**C**) were examined. Values are presented as means ± SD (*n* = 3–5). ^abc^ Values are significantly different from the other groups as determined by Tukey’s test (*p* < 0.05).

**Table 1 plants-10-02579-t001:** pH, Flavonoids, total polyphenols and crude protein levels of BBSL and BBSLP.

	pH Value	Flavonoids	Total Polyphenols	Crude Protein
BBSL *	5.84 ± 0.12	11.5 ± 1.5 mg RUE mL^−1^	3.83 ± 0.3 mg GAE mL^−1^	3.5 ± 0.8 mg mL^−1^
BBSLP	-	0.1 ± 0.01 mg RUE mg^−1^	0.03 ± 0.01 mg GAE mg^−1^	0.31 ± 0.04 mg mg^−1^

* BBSL: black bean steamed liquid; BBSLP: black bean steamed liquid lyophilized product; GAE: gallic acid equivalents; RUE: rutin equivalents. Values are presented as means ± SD (*n* = 3–5).

## Data Availability

The datasets used and/or analyzed during the present study are available from the corresponding author on reasonable request.

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
