# Peer review of "By-Products of the Black Soybean Sauce Manufacturing Process as Potential Antioxidant and Anti-Inflammatory Materials for Use as Functional Foods"

_plants, 2021, doi:10.3390/plants10122579_

Round 1
Reviewer 1 Report
L 58 correct „phenolicn acids”
L 434-435 Why 1X Folin Ciocalteu reagent was added to measure proteins content??
There are numerous discrepancies in the description of the antioxidant activity methods and the results presented in the Figure 1. The authors claim that vit C or Trolox was used as a control but there is also EDTA in the Fig 1B. Figure 1 caption include information about ferrous ion chelation but there are no results and protocol for this determination. Moreover, the section 4.4. In vitro antioxidant ability of BBSLP is poorly described.
The authors claim that high antioxidative properties results from the content of flavonoids and polyphenols. However total polyphenols content was only 0,03 mg GAE/mg and flavonoids 0,1 mg RUE/mg. These are rather trace amounts. In my opinion this study needs to be extended in terms of chemical composition analysis of BBSLP.owever, total polyphenols conH
Author Response
We are grateful for the valuable advice and comments of the reviewers. The following are our point-by-point responses to the comments.
L 58 correct „phenolicn acids”
Thanks for this reminder. We were correct it into "phenolic" on page 2, line 61.
L 434-435 Why 1X Folin Ciocalteu reagent was added to measure proteins content??
Folin-Ciocalteu reagent is also called Folin-Ciocalteu’s phenol reagent. It is developed by Foli and Ciocalteu in 1927 for protein determination (Folin and Ciocalteu, 1927).
Because this Folin-Ciocalteu’s reagent consists of a mixture of sodium molybdate, sodium tungstate and other reagents. This reaction mixture will reactive with the phenol group of protein, then present a blue color which absorbs at 600-765 nm. The blue color is due to a complexed Mo(V) species (Singleton and Rossi, 1965). Later, for great sensitivity and the simplicity of procedure possible, the measured sample will be pre-reacted with "Lowry reagent", which includes 2 % N2CO3 in 0.10 N NaOH, 0.5% CuS04.5H20 in 1 % potassium tartrate, alkaline copper solution and carbonate-copper solution. Then, the reacted subtract were furthermore reactive with the Folin-Ciocalteu’s reagent. This modified method of protein content measure was called the Folin-Lowry assay or Lowry assay is used for the quantitation of proteins (Lowry et al., 1951).
- Folin, O. and Ciocalteu, V. Tyrosine and tryptophan determination in proteins. J. Biol Chem. 1927;73:649–673. [Google Scholar]
- Singleton V, Rossi J. Colorimetry of total phenolics with phosphomolybdic-phosphotungstic acid reagents. Am. J. Enol. Vitic. 1965;16:144–158. [Google Scholar]
- Prior R, Wu X, Schaich K. Standardized methods for the determination of antioxidant capacity and phenolics in foods and dietary supplements. J. Agric. Food Chem. 2005;53:4290–4302. [PubMed] [Google Scholar]
- Lowry O, Rosebrough J, Farr A, Randall R. Protein measurement with the Folin-Phenol reagents. J. Biol. Chem. 1951;193:265–275. [PubMed] [Google Scholar]
There are numerous discrepancies in the description of the antioxidant activity methods and the results presented in the Figure 1. The authors claim that vit C or Trolox was used as a control but there is also EDTA in the Fig 1B. Figure 1 caption include information about ferrous ion chelation but there are no results and protocol for this determination. Moreover, the section 4.4. In vitro antioxidant ability of BBSLP is poorly described.
Thank you for this comment. We have revised the figure 1 caption on page 3, lines 144-145.
We have added more detail described the Section 4.4. In vitro antioxidant ability of BBSLP on page 10, line 495-500; page 10, lines 504-511; page 10, lines 514-519, respectively.
The authors claim that high antioxidative properties results from the content of flavonoids and polyphenols. However total polyphenols content was only 0,03 mg GAE/mg and flavonoids 0,1 mg RUE/mg. These are rather trace amounts. In my opinion this study needs to be extended in terms of chemical composition analysis of BBSLP.owever, total polyphenols conH
Thank you for this comments. We according to you and another reviewer's comments, we have rewritten this paragraph on page 7, lines 362 to page 8, lines371. And, we added a sentence on page 8 lines 367-371. As follows" However, polyphenols or the other components of BBSL and BBSLP whether playing a major or important role in antioxidation or not, are still need furthermore composition analysis and investigation.
Reviewer 2 Report
Author of the Manuscript ID: plants-1444009
Title: Origin and function of structural diversity in the plant specialized metabolome performed an important work, useful for scientific society. There are some issues and errors where the author can concentrate to improve the manuscript prior to its publication.
Comments:
Line 38 and 49: Similar sentences
Line 51 -54: Give citations.
Line 84: Define BBSL. Also, mention in brief about the process of making BBSL in the text with suitable references.
Material and methods, Line 405: Give manufacturing company name and address and model of instruments you used. “rotary vacuum dryer”
Line 433: Give manufacturing company name and address “Lowry Reagent (Sigma-Aldrich Co.)”
Line 443: Who did the analysis of the work? Shimada? Not you?
Line 460: What induced?........“then induced”
Line 467 and 468, 477, 481, 499, 502: Need correction
“Denizot and Lang (1986) [44]”……….. Lowry et al. (1951) [40]………….. Fraga et al. (1988) [45].
Line 480: Denizot and 467 Lang (1986) [44], Hsieh et al. (2020) [47].,, Lowry et al. (1951) [40].
Line480: Give manufacturing company name and address
“(Bio-Tek Instruments)”.
Results: 93, 94, 97: Replace “is” by “was”
And “are” by “were”
Line: 99
“2.2. BBSLP showed antioxidant effects in vitro” replace by “2.2. BBSLP showed antioxidant effects in in vitro conditions”
Table1: Keep the unit of measurement as “Flavonoids (mg RUW mL-1). Make a similar correction throughout the text. Statistical works are missing.
Figure 1,2,3,4,5 and 6: Revise fig 1. means±SD is not correct.
Line 173: need a correction
“TBARS significantly increased TBARS levels by 229%”
Figure 4: indicate abc in the fig 4.
“abcValues are significantly different from the other groups as determined by Tukey’s 244 test (p<0.05)”
Line 2510252: make a new sentence
“however, the IL-1β levels in RAW264.7 macrophages significantly decreased after treatment with 0.5 or 1 μg/mL BBSLP combined with LPS (54.9 ± 4.7, 54.1 ± 12.4, and 69.1 ± 1.8%, respectively) compared with those after induction 252 with LPS alone (100%)……”
Line : 382, 384
Check the format
“Bao et al. (2018)”, Singh et al. (2019)
Discuss molecular mechanisms to support the results and give some recent citations related to the present work.
Author Response
We are grateful for the valuable advice and comments of the reviewers. The following are our point-by-point responses to the comments.
Line 38 and 49: Similar sentences
Thank you for the reviewer's comment. We have revised the on page 1, lines 40-41 and page 2, lines 51-52.
Line 51 -54: Give citations.
Thank you for the reviewer's comment. We have added references, including references no.4 and 5, on page 2, lines 54 and 56.
Line 84: Define BBSL. Also, mention in brief about the process of making BBSL in the text with suitable references.
The first mention of BBSL is on page 2 line 62. It has a full name as "black soybean steamed liquid", and it is abbreviated as BBSL.
On page 9, lines 451453, we have added the process of making BBSL as following“BBSL is collected from the soybean sauce manufacturing process. After fresh black beans are washed and steamed by 120 ℃ streams in a closed steam tank for 1 h. BBSL were directly collected as experimental materials.”
Material and methods, Line 405: Give manufacturing company name and address and model of instruments you used. “rotary vacuum dryer”
We have revised and added these as follow “…rotary evaporator (N-1110, Tokyo Rikakikai Co., Ltd., Tokyo, Japan) and then dried in a freeze dryer (Labconco Freezone 4.5, Kansas City, MO, USA)…”. on page 9, line 457-459.
Line 433: Give manufacturing company name and address “Lowry Reagent (Sigma-Aldrich Co.)”
We have added it to the manuscript on page 9, line 446. And, we have rechecked and revised the whole manuscript throughout.
Line 443: Who did the analysis of the work? Shimada? Not you?
Thank you for this reminder. It is our mistake. We do this test by ourselves. We have revised on page 10, line 503.
Line 460: What induced?........“then induced”
Thank you for this reminder. It is our mistake. The inducer is LPS in this study. We have revised on page 11, line 534.
Line 467 and 468, 477, 481, 499, 502: Need correction
“Denizot and Lang (1986) [44]”……….. Lowry et al. (1951) [40]………….. Fraga et al. (1988) [45].
Line 480: Denizot and 467 Lang (1986) [44], Hsieh et al. (2020) [47].,, Lowry et al. (1951) [40].
Thank you for this reminder. We have revised these mistakes. And, we have rechecked and revised the whole manuscript throughout.
Line480: Give manufacturing company name and address
“(Bio-Tek Instruments)”.
Thank you for the reviewer’s reminder. We have added the manufacturing company name and address on page 11, line 551. And, we have rechecked and revised the whole manuscript throughout.
Results: 93, 94, 97: Replace “is” by “was”
And “are” by “were”
Thank you for the reviewer’s comments. We have revised these mistakes on page 2, lines 99,101 and page 3, line 105.
Line: 99
“2.2. BBSLP showed antioxidant effects in vitro” replace by “2.2. BBSLP showed antioxidant effects in in vitro conditions”
Thank you for this comments. We have revised these mistakes on page 3, line 107.
Table1: Keep the unit of measurement as “Flavonoids (mg RUW mL-1). Make a similar correction throughout the text. Statistical works are missing.
Thank you for the reviewer’s comments. We have been revised the unit as “RUE mL-1”, ”mg GAE mL-1、and mg mL-1 in Table 1 on page 3.
This table showed the pH value, and the levels of flavonoids, total polyphenols and crude protein levels of BBSL and BBSLP. It does not describe and discuss the comparative content between BBSL and BBSLP in these compositions and characteristics. Because BBSL is a liquid sample and BBSLP is a powder sample. So, they are no statistical analysis of these data. But they all are three or more separate experimental test results. Values are presented as means±SD (n=3–5).
Figure 1,2,3,4,5 and 6: Revise fig 1. means±SD is not correct.
Thank you for the reviewer's reminder. It is our mistake. We have replaced figure 1B.
Line 173: need a correction
“TBARS significantly increased TBARS levels by 229%”
Thank you for this reminder. We have revised as “…the TBARS level was significantly increased by 229% (p<0.05)…” on page4, lines 184-185.
Figure 4: indicate abc in the fig 4.
“abcValues are significantly different from the other groups as determined by Tukey’s 244 test (p<0.05)”
I am sorry! We can not understand the reviewer’s comment. We have been indicated the "abc" on the top of histogram and described in figure 4 caption.
Line 2510252: make a new sentence
“however, the IL-1β levels in RAW264.7 macrophages significantly decreased after treatment with 0.5 or 1 μg/mL BBSLP combined with LPS (54.9 ± 4.7, 54.1 ± 12.4, and 69.1 ± 1.8%, respectively) compared with those after induction 252 with LPS alone (100%)……”
Thank you for this reminder. We have rewritten as "... when RAW264.7 macrophages were treated with 0.5 or 1 μg/mL BBSLP combined with LPS, the IL-1β levels were 59.9 ± 8.7% and 32.0 ± 8.7%, they were significantly lower than the LPS induction alone group (100%)" on page 6, lines 272-277.
Line : 382, 384
Check the format
“Bao et al. (2018)”, Singh et al. (2019)
Thank you for this reminder. We have revised into “Bao et al. [30]”, Singh et al. [31] on page 8, lines 405 and 407. And, we have rechecked and revised the whole manuscript throughout.
Discuss molecular mechanisms to support the results and give some recent citations related to the present work.
According to reviewer comments, we citation and replaced some recent references.
Reviewer 3 Report
In my opinion, the sentences are too wordy throughout the manuscript. Sentences should be simple. Needs to revise abstract and introduction.
The authors have used the old references in many places. Try to replace them with new ones.
Section 4.1. The authors have mentioned, “BBSL was a gift from Ta-Tung Soya Sauce Co.” This is not the right way to explain the material. Add all the required details.
Section 4.8 and 4.9 needs to be expanded to reproduce the method.
Labeling of figures is not clear to see properly. Replace with more clear.
The conclusion needs to be rewritten.
Author Response
We are grateful for the valuable advice and comments of the reviewers. The following are our point-by-point responses to the comments.
In my opinion, the sentences are too wordy throughout the manuscript. Sentences should be simple. Needs to revise abstract and introduction.
Thank you for the reviewer's comment. To more simply and focus on this manuscript, the revised paragraph was included the abstract section, on page 1 line 19, and the introduction section, page 2, lines 82-96.
The authors have used the old references in many places. Try to replace them with new ones.
According to reviewer comments, we replaced many new references.
Section 4.1. The authors have mentioned, “BBSL was a gift from Ta-Tung Soya Sauce Co.” This is not the right way to explain the material. Add all the required details.
Thank you for the reviewer's comments. We have added the preparation process of BBSL on page 9, lines 451-453.
Section 4.8 and 4.9 needs to be expanded to reproduce the method.
Thank you for the reviewer's comments. We have added the more detailed measures protocol of NO, PGE2, cytokines in sections 4.8 and 4.9.
Labeling of figures is not clear to see properly. Replace with more clear.
Thanks for the reviewer's reminder. We have checked and enhanced all figure quality.
The conclusion needs to be rewritten.
Thanks for the reviewer's reminder. We have rewritten the conclusion on page 12, lines 604-612.
Reviewer 4 Report
The paper deals with By-Products of the Black Soybean Sauce Manufacturing Process as Potential Antioxidant and Anti-inflammatory Materials for Use as Functional Foods. Good Results part. pLease see below my suggestions:
L81-89. Aim of the study is well defined but please always make it a separate last paragraph of the Introduction. It is more easier visible.
As the Instructions for authors state: Acronyms/Abbreviations/Initialisms should be defined the first time they appear in each of three sections: the abstract; the main text; the first figure or table. When defined for the first time, the acronym/abbreviation/initialism should be added in parentheses after the written-out form." Please proceed accordingly and revise the entire manuscript in this regard.
L54-57. An important disorder that must be mention here (and it was forgotten) is menopause with its bone resorption - please check https://doi.org/10.3390/jcm7100297
Maybe the authors can improve the quality of the Figures provided.
L356-357. It must be mentioned the fact that polyphenols and isoflavones actions are strictly related to their chemical structure. The idea is very well developed in https://doi.org/10.37358/RC.19.9.7497
Weakness and strengths of this study would be also relevant to be presented at the final of Discussion section.
Please develop better/more the Conclusions part.
References must be set according to the journal's Instruction for authors. Please check https://www.mdpi.com/journal/ijerph/instructions ; EndNoting them (if you can) and choosing the MDPI style is very useful . More than half cited references are older that 10 to 50 years ago! I suggest to up-date some of them as, the literature is plenty of published data in the topic debated.
Author Response
We are grateful for the valuable advice and comments of the reviewers. The following are our point-by-point responses to the comments.
L81-89. Aim of the study is well defined but please always make it a separate last paragraph of the Introduction. It is more easier visible.
Thank you for the reviewer's comment. We have rewritten and added the last paragraph of the "Introduction" section on page 2, lines 86-87.
As the Instructions for authors state: Acronyms/Abbreviations/Initialisms should be defined the first time they appear in each of three sections: the abstract; the main text; the first figure or table. When defined for the first time, the acronym/abbreviation/initialism should be added in parentheses after the written-out form." Please proceed accordingly and revise the entire manuscript in this regard.
Thank you for this reminder. We have checked, revised and added the whole manuscript throughout.
L54-57. An important disorder that must be mention here (and it was forgotten) is menopause with its bone resorption - please check https://doi.org/10.3390/jcm7100297
Thank you for the reviewer's comment. According to this reference, we have added and cited the function of black soybean on page 2, line 58.
Maybe the authors can improve the quality of the Figures provided.
Thanks for the reviewer's reminder. We have checked and enhanced all figure quality.
L356-357. It must be mentioned the fact that polyphenols and isoflavones actions are strictly related to their chemical structure. The idea is very well developed in https://doi.org/10.37358/RC.19.9.7497
Thank you for the reviewer's comment. We have rewritten and added and cited this reference in this paragraph on page 7, lines 362-366.
Weakness and strengths of this study would be also relevant to be presented at the final of Discussion section.
To present the weakness, strengths and future work of this study. We have added and revised a paragraph on page 9 lines 438-446. It is as follows.
In the present study, BBSLP showed preliminary potential on antioxidant and anti-inflammatory effects in vitro. However, the molecular mechanisms of the physiological effects require further study in animal models or human clinical trials. On the other hand, how does BBSLP apply in function food? When BBSLP be developed into a functional food material, a functional assessment of this product is needed. The above questions are an important issue about BBSLP application.
Please develop better/more the Conclusions part.
We have rewritten the conclusion on page 12, line 604-612.
References must be set according to the journal's Instruction for authors. Please check https://www.mdpi.com/journal/ijerph/instructions ; EndNoting them (if you can) and choosing the MDPI style is very useful . More than half cited references are older that 10 to 50 years ago! I suggest to up-date some of them as, the literature is plenty of published data in the topic debated.
Thank you for this reminder. We replaced many new references and recheck the reference form on this manuscript.
Round 2
Reviewer 1 Report
I accept introduced corrections and thanks for explanations. I recommend to accept the manuscript for publication.
Reviewer 2 Report
Author of the manuscript (ID plants-1460037) responded and revised the research article carefully and answered all the queries. Now the manuscript can be accepted for its publication.